# Biocontrol Potential of Endophytic *Bacillus velezensis* LSR7 Against Rubber Red Root Rot Disease

**DOI:** 10.3390/jof10120849

**Published:** 2024-12-09

**Authors:** Xiangjia Meng, Haibin Cai, Youhong Luo, Xinyang Zhao, Yongwei Fu, Lifang Zou, Yi Zhou, Min Tu

**Affiliations:** 1National Key Laboratory for Tropical Crop Breeding, Rubber Research Institute, Chinese Academy of Tropical Agricultural Sciences, Sanya 572024, China; 2023730089@yangtzeu.edu.cn (X.M.); haibin_cai@163.com (H.C.); 2021710770@yangtzeu.edu.cn (Y.L.); xyzhao1219@163.com (X.Z.); fu65227795@163.com (Y.F.); 2School of Agriculture, Yangtze University, Jingzhou 434000, China; 3Shanghai Collaborative Innovation Center of Agri-Seeds, School of Agriculture and Biology, Shanghai Jiao Tong University, Shanghai 200240, China; zoulifang202018@sjtu.edu.cn; 4Sanya Research Institute, Chinese Academy of Tropical Agricultural Sciences, Sanya 572020, China

**Keywords:** *Bacillus velezensis*, rubber red root rot diseases, *Ganoderma pseudoferreum*, biological control

## Abstract

To obtain an effective bacterial biocontrol strain against the fungal pathogen *Ganoderma pseudoferreum*, causing rubber tree red root rot disease, healthy rubber tree tissue from Baisha County, Hainan Province, was selected as the isolation source, and bacterial strains with strong antifungal effects against *G*. *pseudoferreum* were screened. The strain was identified by molecular biology, in vitro root segment tests, pot growth promotion tests, and genome detection. The strain was further evaluated by biological function tests, genome annotation analysis, and plant defense-related enzyme activity detection. The results show that strain LSR7 had good antagonistic effects against *G. pseudoferreum*, and the inhibition rate reached 88.49%. The strain LSR7 was identified as *Bacillus velezensis* by genome sequencing. In a greenhouse environment, LSR7 prevents and treats red root rot disease in rubber trees and promotes the growth of rubber tree seedlings. LSR7 secreted cell wall hydrolases (protease, glucanase, and cellulase), amylases, and siderophores. LSR7 also formed biofilms, facilitating plant colonization. Genome prediction showed that LSR7 secreted multiple antifungal lipopeptides. LSR7 enhanced rubber tree resistance to *G. pseudoferreum* by increasing the activity of defense enzymes. *Bacillus velezensis* LSR7 has biocontrol potential and is a candidate strain for controlling red root rot disease in rubber trees.

## 1. Introduction

Natural rubber (NR) has unique properties, including wear resistance, dynamic mechanical properties, flexibility, and strength, which cannot be fully replaced by synthetic rubber (SR) [1,2]. *Hevea brasiliensis* is the only important cash crop that produces natural rubber on a large scale and is cultivated commercially in tropical regions globally [3]. Owing to global industrial and commercial demand, natural rubber forests have gradually developed into artificially cultivated rubber forests. The absence of genetic diversity and prevalence of monoculture practices render rubber trees highly susceptible to pathogens, thereby increasing their susceptibility to various diseases [4]. Root and leaf diseases, such as powdery mildew, *Colletotrichum* leaf disease, red root rot disease etc. are major threats to rubber trees [4].

Red root rot disease, caused by *Ganoderma pseudoferreum* (Wakef.) Over. et Steinm, is a widespread, economically damaging root-borne infectious disease in Chinese rubber tree plantations [5,6]. Current practices include chemical pesticide irrigation before disease onset, trench digging isolating the diseased plant after disease onset, and controlling disease transmission path by removing affected plants [7]. However, digging trenches for isolation is time-consuming and inefficient, and long-term pesticide irrigation causes irreversible ecological damage and pathogen resistance [8,9,10]. To overcome the disadvantages of traditional control measures, the use of living microorganisms and their metabolites to inhibit the spread of plant diseases has become a new research focus globally. Biocontrol agents (BCAs) are used to prevent and treat rubber tree diseases. *Bacillus velezensis* effectively controls powdery mildew and *Colletotrichum* leaf disease in rubber trees [11,12]. The co-inoculation of arbuscular mycorrhizal fungus, *Enterobacter* sp. UPMSSB7, and silicon suppress *Rigidoporus microporus* and reduce the severity of rubber tree white root rot disease, promoting rubber seedling growth [13]. However, research on microorganisms to control red root rot in rubber trees is limited. Therefore, obtaining a bacterial strain that can sustainably and effectively control of rubber tree red root rot is crucial for the biological of rubber tree management.

Plant growth-promoting bacteria (PGPB) are used for the biological control of plant pathogens [9]. PGPB can inhabit the rhizosphere, epidermis, and interior of plants [14]. Endophytic bacteria, compared to non-endophytes, may enter and colonize their original plant hosts more easily, as they are adapted to co-exist with the plant host without triggering defense mechanisms. Therefore, these bacteria are termed PGPEB [15,16,17]. Endophytic bacteria in plants can inhibit or kill pathogens by producing lytic enzymes (cellulase, protease, and chitinase) and secondary metabolites. They also limit the development of pathogens by competing for living space within plant tissues, while, at the same time, can induce disease resistance in host plants [18]. Endophytes can provide a nitrogen source for plant growth through nitrogen fixation and can also produce IAA and siderophores to promote plant growth [15,19,20]. PGPB are increasingly recognized as biocontrol agents and plant growth promoters. For example, volatile organic compounds by endophytic *Bacillus safensis* Har267 and *B. aerius* Kh867 reduced *Ralstonia solanacearum* R32 aggregates, motility, biofilm formation, and root tissue attachment, causing bacterial potato wilt [21]. *Bacillus* sp. E2 from municipal wastewater-irrigated cucumber roots antagonized *Fusarium oxysporum*, producing IAA, hydrogen cyanide, and phosphate solubilization, while alleviating heavy metal stress [22]. Several *Bacillus* spp. isolated from rice leaves have antifungal activity against *Pyricularia oryzae*, increasing plant disease resistance by stimulating the relevant defense enzymes and genes [23]. PGPB’s antagonism to plant pathogens makes them an excellent choice to replace pesticides, etc., and they should be applied in environmentally friendly agricultural practices [24].

Endophytic microorganisms can control rubber tree diseases. For example, endophytic *Trichoderma* isolated from healthy rubber tree leaves effectively prevents *Phytophthora* leaf fall disease under greenhouse and field conditions [25]. There are few reports on the biocontrol of rubber root rot disease by endophytic microorganisms. Therefore, this study aimed to (1) screen for BCAs with antagonistic effects on *G. pseudoferreum* in endophytic bacteria isolated from healthy rubber tree tissues; (2) determine their biocontrol activity in controlling rubber red root rot disease under greenhouse conditions; (3) identify mechanisms of BCA antagonism against *G. pseudoferreum*, including hydrolase, siderophores, and antifungal secondary metabolite production; and (4) evaluate induced disease resistance in rubber trees using BCA by measuring defense-related enzyme activity levels.

## 2. Materials and Methods

### 2.1. Endophytic Bacterium Isolation from Rubber Tree

Endophytic bacteria were isolated from rubber trees using tissue separation and dilution plating methods. Healthy rubber tree (Longjiang farm, Baisha County, Hainan Province, China) leaves, roots, and stems were washed with clean water to remove surface impurities, dried with sterile filter paper, and rinsed three times with sterile water in an ultra-clean bench. Moisture was absorbed using sterile filter paper and soaked in 2% NaClO solution for 5 min. After removal, the samples were soaked in 75% ethanol for 1 min for surface disinfection. Thereafter, the samples were rinsed three times with sterile water, and moisture was absorbed using sterile paper. Plant tissues were ground using a sterile mortar, and the ground tissue fluid was collected for 10-fold dilution and plating. The plant tissue was ground with sterile mortar; the tissue liquid collected after grinding was diluted 10-fold with sterile water and plated in nutrient agar (NA) plates. When bacterial colonies appeared, a single colony was streaked onto a nutrient agar (NA) plate for purification using an inoculation needle. After uniform numbering, purified colonies were stored at 4 °C in a refrigerator. To simultaneously confirm the isolated bacteria as endophytic fungi of rubber trees, sterile water from the final rinse was used as a control for plating. If no bacteria grew on the control plate, the isolated bacteria were confirmed as endophytic [26].

### 2.2. Screening of Antagonistic Bacteria from Rubber Tree

Endophytic bacteria with antagonistic effects against *G. pseudoferreum* were screened using the plate-confrontation culture method. The fungal pathogen CATAS-RRI-Gp-01 was isolated from the Experimental Station of the Chinese Academy of Tropical Agricultural Sciences (Hainan, Danzhou, China) in the early stage, and ITS, nrDNA-LSU, and nrDNA-SSU sequences (GenBank accession No: KX454313, KX454418 and KX454439) were identified as *G. pseudoferreum*. Nutrient broth (NB) medium (100 mL) was inoculated with isolated endophytic bacteria and cultured in a constant temperature shaker (180 rpm, 28 °C) for 2 d. A 5 mm mycelial plug of activated *G. pseudoferreum* was placed in the center of a flour dextrose agar (FDA) (30 g flour, 15 g dextrose, 12 g agar, 1 L distilled water, pH = 7.0) plate. Holes were punched at a right triangle distance of 25 mm from the *G. pseudoferreum* mycelial plug using a 5 mm puncher, and 100 µL of endophytic bacterial culture was added to each hole. Plates inoculated with equal amounts of NB were used as controls, and each treatment was repeated thrice (n = 15). The plates were incubated at 28 °C for 7 d in a constant temperature incubator (Thermo Fisher Scientific, Shanghai, China). The colony diameter of the red root fungus was measured using a ruler, and its relative inhibition rate was calculated as follows [27]:Inhibition rate = [(Colony diameter of control *G. pseudoferreum* − Colony diameter of treated *G. pseudoferreum*)/(Colony diameter of control *G. pseudoferreum* − 5.0 mm)] × 100%

### 2.3. Identification of LSR7

The LSR7 strain, which showed the strongest inhibitory activity against *G. pseudoferreum*, was selected for subsequent experiments. To preliminarily identify highly efficient antagonistic endophytic bacteria LSR7, the total DNA of LSR7 was extracted using a bacterial DNA kit (Omega Bio-tek, Norcross, GA, USA). The extraction method was performed according to the manufacturer’s instructions. The *16S* rDNA and *gyrB* [28,29] genes of bacterial DNA were amplified via PCR using the amplification system and procedure as described in Appendix A. Five microliters of the PCR product were subjected to gel electrophoresis to confirm amplification. The confirmed gene fragment was sequenced and analyzed by BGI Bioengineering Co., Ltd. (Shanghai, China). The sequencing results were compared to the NCBI database (https://www.ncbi.nlm.nih.gov/) using BLAST. Phylogenetic trees were constructed using the neighbor-joining method in MEGA 7.0, with bootstrap values derived from 1000 replicates.

### 2.4. Effect of LSR7 on Rubber Red Root Rot Disease

Healthy roots from rubber tree clone GT1 (1.3–1.6 cm in diameter) were selected and cut into root segments of approximately 8 cm in length. The root segments were surface-sterilized with 75% ethanol and 2.5% sodium hypochlorite. The LSR7 strain was inoculated in NB and incubated at 27 °C with shaking at 140 rpm for 2 d. The bacterial culture was adjusted to concentrations of 10^8^, 10^7^, and 10^6^ CFU/mL, recorded as P_3_, P_2_, and P_1_, respectively. Root segments were immersed in LSR7 bacterial suspensions at the stipulated concentrations for 1 h; NB served as the negative control (CK), and tridemorph as the positive control. A 2 mm × 2 mm wound was made in the middle epidermis of each root segment with a scalpel, and a 2 mm diameter *G. pseudoferreum* fungal disc was inoculated onto the wound. Both ends of the root segment were moistened with absorbent cotton, and all segments were placed in a humid box and cultured at 28 °C for 7 d (light/dark cycle of 0:24 h). There were 20 root segments per treatment, and the experiment was repeated in triplicate. The relative lesion area was calculated using Image-Pro Plus 6.0 software.
Control or preventive effect = (1 − relative lesion area in the CK group/relative lesion area in the treatment group) × 100%

The effect of LSR7 on rubber tree red root rot disease was determined using real-time quantitative (qPCR). Total RNA (DNase digestion was used to remove DNA from the sample without RNase) was extracted from the mycelia of *G. pseudoferreum* on each wound in the root segments that were using the RNA Prep Pure Plant Kit (TIANGEN, Beijing, China). cDNA was synthesized using the ToloScript ALL-in-one reverse transcriptomeEasyMix qPCR kit (TOLOBIO, Irvine, CA, USA). cDNA was amplified via qPCR using *ACT* (target gene) and *18S* rRNA (internal reference) as the internal reference genes, with the amplification system and program listed in Appendix A. Thereafter, the cycle threshold was exported using Origin 2021, and the relative gene expression levels were calculated using the 2^−ΔΔCT^ method [30]. The group with NB treatment but without *G. pseudoferreum* inoculation was used as the control (CK_0_), and the other treatment groups were the same as previously mentioned.
Control or preventive effect = (1 − relative gene expression in the CK group/relative gene expression in the treatment group) × 100%

### 2.5. Effect of LSR7 on Rubber Tree Seedling Growth

The LSR7 bacterial culture was prepared according to the method described in Section 2.4, and the culture solution concentration was adjusted and mixed with four sterile substrate soils to achieve LSR7 concentrations of 10^9^, 10^8^, 10^7^, and 10^6^ CFU/mL as four treatments, denoted as Q_4_, Q_3_, Q_2_, and Q_1_, respectively. Simultaneously, NB served as the control. Seedlings of the rubber tree clone GT1 (height 20 cm) were transplanted into plastic pots (diameter 12 cm × height 10 cm) filled with substrate soil and cultivated in a greenhouse at 28 °C under a light/dark cycle of 16:8 h. Each treatment included 20 rubber tree seedlings, and the experiment was performed in triplicate. Growth parameters, including seedling height, root length, root dry weight, stem dry weight, chlorophyll content, and number of root systems of rubber seedlings were measured after 30 d of treatment. 

### 2.6. Analysis of the Characteristics of Endophytic Bacteria

#### 2.6.1. Siderophore Production Determination

To test the iron carrier activity of LSR7, Chrome Azurol S agar media were prepared according to the method described by Shin et al. [31]. Two holes were punched 25 mm distance from the center of the plate using a 5 mm diameter puncher. One hole was inoculated with 100 μL of LSR7 bacterial culture (48 h) and the other hole served as the control with NB medium. Each treatment was repeated thrice. All Petri dishes were incubated at 28 °C for 3 d. Thereafter, clear zones around the colonies were measured. 

#### 2.6.2. Detection of LSR7 Extracellular Enzymes

To test the protease, cellulase, chitinase, amylase, and β-1, 3-glucanase activities of LSR7, skim milk lactose [32], carboxymethyl cellulose [33], colloidal chitosan [34], and amylase detection [35] agar media were prepared according to the methods described by previous studies. The β-1, 3-glucanase detection agar medium was prepared using 5 g casein, 2.5 g yeast extract, 1 g glucose, 1% carboxymethyl cellulose, 15 g agar powder, and 1 L distilled water (pH 7.0). LSR7 was inoculated in five types of test culture medium plates according to the method in Section 2.6.1, with NB medium as the control. Each treatment was repeated thrice. All plates were incubated at 28 °C for 3 d. The carboxymethyl cellulose agar medium was stained with Congo red (1 mg/mL) for 15 min and subsequently rinsed with NaCl (1 mol/L) for 10 min; clear circles around the colonies indicated cellulase activity. After submerging the amylase detection medium in iodine solution for 10 min, if a clear annular band is observed, it indicates that the strain produces amylase. The other enzyme detection media were directly observed for the presence or absence of clear circles around the colonies.

#### 2.6.3. Detection of LSR7 Biofilm Formation

The ability of LSR7 to form biofilms was tested using glass test tubes. The strain was inoculated in NB medium and cultured at 28 °C, 130 rpm for 1 d. The bacterial suspension was diluted to different concentrations with NB medium (OD_600_ = 0, 0.2, 0.4, 0.6, 0.8, and 1.0), which had 4 mL injected into each test tube; this was repeated in triplicate for each concentration and incubated at 28 °C for 2 d. Five milliliters of crystal violet solution (0.1%, *w*/*v*) were added to each test tube, left at 28 °C for 30 min. Thereafter, the liquid was poured out and the test tube was washed with sterile water; if a purple ring mark appeared on the wall of the test tube, it indicated that a biofilm was formed. The biofilm was dissolved in 5 mL of 95% ethanol and the absorbance of the solution at 570 nm was determined using a UV vis spectrophotometer (UV-5100B, Shanghai, China) to determine the amount of biofilm produced. The initial absorbance value was determined with nb as the control [36].

### 2.7. LSR7 Genomic Sequencing and Annotation

LSR7 was inoculated in NB and cultured at 28 °C, with an oscillation rate of 130 rpm min^−1^ until it reached the logarithmic stage. The bacterial LSR7 solution was centrifuged at 4000 rpm at 4 °C for 15 min. The resulting precipitate was collected and washed thrice using sterile water. Supernatants were discarded afterward. Liquid nitrogen was used to freeze the remaining bacterial sediments and send them to Benagen in Wuhan, Hubei Province, China. Utilizing next-generation sequencing (NGS), based on the Illumina NovaSeq 6000 (Illumina, San Diego, CA, USA) sequencing platform, and simultaneously employing third-generation single-molecule sequencing technology, based on the PromethION (Oxford Nanopore Technologies, Oxford, UK) sequencing platform, these libraries were sequenced respectively. After filtering the second-generation raw sequencing data with fastp (Version: 0.23.2), effective sequencing data were obtained. The filtered reads were assembled using the Unicycler (Version: 0.5.0) software [37], and raw data from Nanopore sequencing were in the fast5 format, which were converted into the fastq format after base calling through GUPPY. Then, the raw sequencing data were subjected to quality filtering (Q ≥ 7) and length filtering (length ≥ 1600 bp) to obtain valid data for the subsequent assembly analysis. This preliminary annotation was completed by using the Prokka (Version 1.14.6) software [38], which uses Prodigal, Aragorn, RNAmmer, and Infernal to predict the coding gene, tRNA, rRNA, RNAmmer, and miscRNA, respectively. To obtain comprehensive gene function information, we conducted gene function annotation in three major databases: Kyoto Encyclopedia of Genes and Genomes (KEGG) Pathway, Gene Ontology (GO), and Clusters of Orthologous Groups (COG). The R package circlize [39] was used to integrate the predicted genomic information and draw the genomic cycle map. Secondary metabolite gene clusters were analyzed and predicted using the antiSMASH online tool (https://antismash.secondarymetabolites.org/#!/start; accessed on 19 March 2024). The Genome BLAST Distance Phylogeny approach was used to calculate the phylogeny between LSR7 and homologous strains based on the LSR7 16S rDNA gene. Analyses were conducted and using online websites (https://jspecies.ribohost.com/jspeciesws/#home accessed on 29 June 2024) for average nucleotide identity (ANI) and (https://ggdc.dsmz.de/, accessed on 15 May 2023) to compare for DNA-DNA hybridization (DDH). 

### 2.8. Antifungal Activity of LSR7 Secondary Metabolites

#### 2.8.1. Effect of Temperature and Protease on Bacterial Culture Filtrate Antagonistic Activity

The bacterial culture filtrate (BCF) of strain LSR7 was obtained according to the method described by of Meng et al. [40] for future use. The BCF was processed as follows: (1) BCF was thoroughly mixed with uncured FDA medium at concentrations of 5, 10, and 20% and poured into the Petri dish; (2) the BCF was heated in a water bath (25, 40, 60, 80, and 100 °C) and autoclave sterilizer (ZEALWAY, Los Angeles, CA, USA) at 121 °C for 30 min, and mixed with uncured FDA at a 20% concentration and poured into Petri dishes; (3) BCF was fully mixed with protease K (Solarbio, Beijing, China) at the concentrations of 0, 10, and 100 μg/mL, and then left for 1 h, mixed with FDA at the ratio of 1:5 (*v*/*v*), and poured into Petri dishes. NB of equal-volume BCF in the treatment group was mixed with FDA as the control (CK), with each treatment performed in triplicate (n = 15). *G. pseudoferreum* mycelial plugs were inoculated in the center of the plate and incubated at 28 °C until the control group mycelia covered the entire plate. The inhibition rate was calculated as described in Section 2.2.

#### 2.8.2. Detection of Antifungal Activity in Lipopeptide Metabolites of LSR7 Extracted by Different Solvents

The strain LSR7 was inoculated into NB medium and cultured at 28 °C, 130 rpm for 7 d. The bacterial culture was centrifuged at 10,000 rpm at 25 °C for 30 min, after which the bacterial cells were remove and the supernatant was collected. The supernatant was divided into four equal parts and mixed with equal volumes of ethyl acetate, n-butanol alcohol, petroleum ether, and acetone, shaken in a separating funnel and left to completely the layer before the upper organic phase was collected. The lower aqueous phase was extracted by adding solvent three times consecutively. The organic phases from the three extractions were combined, and the organic solvent was removed using a rotary evaporator (Thermo Fisher Scientific, Wuhan, China) at 60 °C. Subsequently, the crude lipopeptide extract of strain LSR7 was obtained. The crude extract was mixed with FDA to achieve a concentration of 100 ug/mL in the FDA. The aqueous phase after extraction was used as the control, with each treatment performed in triplicate (n = 15). The antifungal activity of LSR7 lipopeptide metabolites by different organic solvents against G. pseudoferreum was calculated using the method described in Section 2.8.1.

### 2.9. Detection of Defense-Related Enzyme Activities in Plants 

The enzyme activity related to the defense mechanisms of LSR7 strain against the rubber tree roots was also detected. Experiments were conducted using the following treatments: rubber tree roots were immersed in Hoagland’s nutrient solution (PHYGENE, Shanghai, China) containing LSR7 bacterial concentrations of 10^9^, 10^8^, 10^7^, and 10^6^ CFU/mL for hydroponic treatment, denoted as M_4_, M_3_, M_2_, and M_1_, respectively. Rubber tree roots were immersed in Hoagland’s nutrient solution without LSR7 as the control treatment. All seedlings were hydroponically grown in a greenhouse at 28 °C with a 16:8 light/dark cycle. Assays of enzyme activity were conducted on roots of the same plant after 1, 2, 3, 4, 5, 6, and 7 days of treatment. These assays measured rubber tree peroxidase (POD), phenylalanine ammonia-lyase (PAL), polyphenol oxidase (PPO), catalase (CAT), and superoxide dismutase (SOD) activity. The extraction of defense enzyme solutions and determination of their activities were performed using a kit (Solarbio, Beijing, China). The detection method was consistent with the instructions in the kit. This experiment was performed in triplicate.

### 2.10. Statistical Analyses

SPSS Version 23 was used to conduct an analysis of variance (ANOVA) of the data. Duncan’s multiple range test was used to compare means of the treatment groups. A significance level of 0.05 (*p* < 0.05) was set for statistical significance.

## 3. Results

### 3.1. The Antagonistic Effect of LSR7 on G. Pseudoferreum and Its Identification

Eighty-five strains were successfully isolated from the tissue of rubber tree. Eight strains of *G. pseudoferreum* were screened by dual-culture experiment, among which LSR7 had the strongest *G. pseudoferreum* effect, and the antifungal effect reached 88.49 ± 9.68% (Figure 1, Table 1). We analyzed the *16S* rDNA and *gyrB* gene sequences of LSR7 against the GenBank database using BLAST. The results in Appendix A indicate that the 16S rDNA gene sequence of LSR7 belongs to the same branch as *Bacillus velezensis* HN2 (PP106361.1) and *B. amyloliquefaciens* NBRC15535 (MK182997.1), and thus, LSR7 was classified under the genus *Bacillus* sp. Moreover, the results in Appendix A show that the *gyrB* sequence of LSR7 belongs to the same branch as *B. velezensis* FZB42, thereby identifying LSR7 as *B. velezensis*. Strain LSR7 was sent to the China Center for Type Culture Collection for identification and preservation, and its storage number was M2024389. The strain culture collection center determined the classification status of strain LSR7 by detecting strain morphological characteristics, 16S rRNA gene sequence, physiological and biochemical characteristics, and whole-genome detection. The results identified by the culture collection center were consistent with ours, that is, the strain LSR7 was *B. velezensis*.

### 3.2. Inhibitory Effect of Strain LSR7 on Rubber Red Root Rot Disease in a Greenhouse

Using *G. pseudoferreum* as a model, we measured the relative lesion (Figure 2B,D) area and gene expression (Figure 2A,C,D) of LSR7 in order to assess how effective it is at controlling rubber tree red root rot. The inhibition rates (relative lesion area) of red root rot disease at LSR7 concentrations of 10^6^ (P_1_), 10^7^ (P_2_), and 10^8^ (P_3_) CFU/mL were 25.68 ± 3.54, 43.58 ± 4.78, and 4.52%, respectively. At a concentration of 10^8^ CFU/mL, LSR7 had no statistically significant difference from tridemorph in its antifungal activity (61.08% ± 5.23). The efficacy (relative gene expression) of LSR7 against red root rot at the concentrations of 10^6^, 10^7^, and 10^8^ CFU/mL was 30.56 ± 6.23, 37.79 ± 4.75, and 55.06 ± 5.86%, respectively, compared with that on the NB medium. LSR7 10^8^ CFU/mL was equivalent to tridemorph (56.84 ± 7.78%) in terms of efficacy. In addition, the results from both methods were consistent.

### 3.3. Effect of LSR7 on Rubber Seedling Growth

LSR7 promoted the growth of rubber tree seedlings (Figure 3). At certain concentrations, LSR7 significantly increased seedling height, root dry weight, stem dry weight, and number of root systems. At the LSR7 concentration of 10^8^ CFU/mL (Q_3_), seedling height, root dry weight, stem dry weight, and number of root systems increased by 21.34%, 33.85%, 61.61%, and 68.75% respectively. At the LSR7 concentration of 10^9^ CFU/mL (Q_4_), these parameters increased by 16.31%, 29.74%, 51.39%, and 43.75% respectively. At the LSR7 concentration of 10^7^ CFU/mL (Q_2_), the increases were 11.03%, 22.05%, 25.08%, and 50.00% respectively. The growth promotion effect of LSR7 on rubber tree seedlings was not significant at 10^6^ CFU/mL (Q_1_), except for the number of root systems. As shown in (Figure 3D,G), LSR7 did not have a significant effect on the root length or chlorophyll content of rubber tree seedlings.

### 3.4. Determination of the Biological Characteristics of Strain LSR7

The medium inoculated with LSR7 showed protease hydrolysis, cellulase hydrolysis, glucan hydrolysis, amylase hydrolysis, and a white halo formed by Fe ions (Figure 4). Strain LSR7 produced protease, cellulase, β-1,3-glucanase, and siderophore functions. Strain LSR7 did not produce a clear chitin zone, indicating it could not produce chitinases. The strain LSR7 formed biofilms (Appendix A). The biofilm formed by strain LSR7 at OD_570_  =  0.8 was significantly stronger than that at OD_570_  =  1.0 or 0.6, *p*  < 0.05.

### 3.5. Genome Sequencing and Annotation of Strain LSR7

The sequencing results of the whole genome of bacteria LSR7 are shown in Appendix A and Appendix A. The genome sequence of strain LSR7 was a circular genome without plasmid sequence. There were 3,899,490 bp in the genome of LSR7 with a GC content of 46.62% and 3711 protein-coding sequences covering 88.47% of its length. The genome contained 86 tRNA, 9 5S rRNA, 9 16S rRNA, and 9 23S rRNA genes. The LSR7 ANI and DDH values exceeded 97.76 and 94.78%, respectively, compared to those of *B. velezensis* FZB4, indicating that LSR7 is a *Bacillus velezensis* strain.The COG, GO, and KEGG databases were used to compare the protein sequences of the predicted genes with the functional databases, and the highest comparison results were selected for annotation. Among them, 3030, 1177, and 2330 genes were functionally annotated in the COG, GO, and KEGG databases, respectively, accounting for 81.65%, 31.72%, and 62.79% of the total genes. In the COG annotations, the most abundant genes were involved in general function prediction (R), amino acid transport and metabolism (E), and transcription (K) (Appendix A). In the GO annotations, 289 genes are associated with biological processes, 673 with cellular components, and 772 with molecular functions (Appendix A). The KEGG analysis revealed that LSR7 has abundant substance metabolic pathways that allow adaptation to the environment (Appendix A).

The anti-SMASH analysis revealed that LSR7 contains 14 gene clusters related to secondary metabolite synthesis. Macrolactin H, bacillaence, fengycin, difficidin, bacillothiazole, bacillibactin, and bacilysin had 100% similarity to known secondary metabolite synthesis gene clusters in the databases, whereas surfactin and plantazolicin exhibited > 85% similarity, indicating that LSR7 synthesized these secondary metabolites (Figure 5 and Appendix A). Additionally, LSR7 had three unknown product gene clusters (Appendix A). A comparative analysis of secondary metabolite gene clusters was performed between LSR7 and four other *Bacillus* strains. Eight secondary metabolite gene clusters encoding surfactin, terpene, bacillaene, fengycin, T3PKS, bacillibactin, and bacilysin were present in all three *Bacillus* strains (Figure 6B). The secondary metabolite gene clusters in LSR7 were highly similar to those of FZB42, except that LSR7 synthesized kijianimicin, which was absent in FZB42. These results suggest that LSR7 may have a biocontrol function similar to that of FZB42 in inhibiting microorganisms, inducing host resistance, and promoting plant growth.

### 3.6. Effect of Temperature and Protease on BCF Antagonistic Activity

The BCF of LSR7 significantly inhibited the growth of *G. pseudoferreum* (Figure 7A,B). The inhibition rates of *G. pseudoferreum* at the LSR7 BCF concentrations of 5, 10, and 20% were 50.61 ± 8.32, 59.86 ± 7.23, and 87.86 ± 6.45, respectively. The antifungal rate of BCF treated with proteinase K at the concentrations of 1, 10, and 100 μg/mL against *G. pseudoferreum* was between 75 and 78%, with no significant difference in antifungal rates among different concentrations (Figure 7C,D). The antifungal rate of BCF against *G. pseudoferreum* after treatment at the temperatures of 25, 40, 60, 80, and 100 °C ranged between 86 and 92%, with no significant difference. When the temperature reached 121 °C, a significant downward trend was observed in the antifungal rate of BCF; however, the antifungal rate could still reach up to 84.89 ± 10.36% (Appendix A). As a result, neither proteinase K nor the temperature affected BCF’s antifungal properties. The secondary antimicrobial metabolites present in BCF consisted predominantly of thermostable lipopeptide compounds.

### 3.7. Detection of Antifungal Activity of LSR7 Lipopeptide Metabolites

The lipopeptide metabolites of LSR7 significantly inhibited the growth of *G. pseudoferreum.* In Figure 8, the lipopeptide metabolites of LSR7 extracted by ethyl acetate, n-butanol alcohol, petroleum ether, and acetone have inhibition rates against *G. pseudoferreum* of 83.25 ± 3.86, 74.63 ± 4.56, 69.85 ± 8.69, and 66.52 ± 8.45%, respectively. The antifungal activity of lipopeptide metabolites extracted with ethyl acetate was the strongest, and the inhibition rate was significantly higher than that of the other three organic solvents.

### 3.8. Effects of LSR7 on the Activity of Rubber Defense-Related Enzymes

Figure 9 shows the changes in POD, PPO, PAL, CAT, and SOD activities in the roots of rubber tree seedlings after LSR7 treatment. The activities of PAL, CAT, and PPO first increased and subsequently decreased. PAL and CAT activities peaked on the 4th day, and PPO activity peaked on the 3rd day. Trends of POD and SOD activities were bimodal, with POD peaking on the 2nd and 4th days, and SOD peaking on the 3rd and 5th days. The concentration of LSR7 at 10^8^ CFU/mL (M_3_) had the most significant effect on the activity of the five enzymes in plants. After treatment with LSR7 at 10^8^ CFU/mL, the activities of PAL, CAT, and POD reached their maximum values of 78.36, 807.63, and 15.87 × 10^3^ U·g^−1^·min^−1^ FW, respectively, on the 4th day. The values were 2.58, 3.11, and 2.32 times that of the control group, respectively. SOD and PPO reached their maximum values of 923.89 and 106.58 U·g^−1^·min^−1^ FW on the 3rd day, at 2.36 and 4.19 times that of the control group, respectively. The activity of related enzymes in rubber tree seedlings increased with the increase in concentrations of LSR7 at 10^6^ (M_1_), 10^7^ (M_2_), and 10^8^ CFU/mL. However, when the concentration of LSR7 was 10^9^ CFU/mL, its effect on the activity of the five enzymes in the plant was not as pronounced as that observed at a concentration of 10^8^ CFU/mL. This suggested that there is a positive correlation between the concentration of LSR7 and the activity of related enzymes in rubber tree seedlings within a certain range.

## 4. Discussion

Red root rot disease significantly hinders the growth of rubber trees and, in severe instances, can lead to their death, thereby causing substantial reductions in natural rubber production. Utilizing beneficial microorganisms represents a safe, efficient, and durable approach for managing red root disease in rubber trees. *Bacillus velezensis* SF305 (isolated from rhizosphere soils) and *Burkholderia cepacia* DHR18 (isolated from rubber tree roots) have a significant preventive effect against rubber tree red root rot disease, with a control efficiency reaching 70% and 64%, respectively [40,41]. However, no reports of controlling rubber tree red root disease using endophytic *Bacillus* in plants exist. *B. velezensis* LSR7 was isolated, characterized, and showed good preventive effects against rubber tree red root rot disease. The antagonistic mechanism of LSR7 against *G. pseudoferreum* was investigated using genome sequencing, siderophores, hydrolases, biofilm formation, extracellular enzyme analysis, and enzyme activity assays. To our knowledge, this is the first time that an endophytic *Bacillus velezensis* has been reported to control rubber tree red root rot. 

The antifungal mechanism study of LSR7 against *G. pseudoferreum* showed that LSR7 can produce protease, β-1,3-glucosidase, and cellulase, unlike *B. velezensis* NKG-2, which produces chitinase that LSR7 cannot [42]. Plant pathogenic fungi rely heavily on cellulose, glucans, and proteins for their cell walls, which play a crucial role in their survival and reproduction [42,43]. The endophytic *Bacillus tequilensis* GYLH001 can inhibit the growth of *Magnaporthe oryzae* by producing cellulase and protease [44]. Therefore, LSR7 may disrupt the structure of the cell wall of plant pathogenic fungi by secreting proteases and cellulases, altering hyphal morphology, and inhibiting *G. pseudoferreum* growth. LSR7 produces amylase and siderophores, consistent with plant growth-promoting bacteria characteristics. Iron carriers secreted by beneficial bacteria promote plant growth under iron-deficient conditions and bind to Fe^3+^, limiting iron availability to pathogenic fungi, hindering their growth [45,46]. Amylases produced by beneficial bacteria may provide absorbable nutrients to plants [47]. Forming biofilms is a key factor in determining the colonization potential of biocontrol bacteria [48,49]. *Bacillus velezensis* FZB42, a model strain, produces biofilms [50]. Notably, LSR7′s biofilm-forming ability suggested its colonization potential in plants.

In this study, the secondary metabolites produced by LSR7 exhibited significant antifungal activity against *G. pseudoferreum*, even after treatment with high temperature and proteinase K, similar to DHR18 [40]. The main components of antifungal secondary metabolites may be non-protein substances. Compared to other organic solvents in the experiment, the lipopeptide metabolites extracted from LSR7 by ethyl acetate have the strongest antifungal activity against *G. pseudoferreum,* similar to *Pseudomonas mosselii* 923 [51]. Predicting synthetic secondary metabolite gene clusters showed that nine of LSR7′s gene clusters shared over 80% similarity with known compounds. The comparative genomic analysis showed LSR7 is more closely related to FZB42 than to SQR9, DSM7, and DSM10, with 100% similarity in seven secondary metabolite gene clusters (macrolactin H, bacillaence, fengycin, difficidin, bacillothiazol, bacillibactin, and bacilysin) with FZB42. These substances are secondary metabolites produced by *B*. *velezensis* and exhibit antibacterial activity [52]. Difficidin, fengycin, and surfatin produced by *B. velezensis* 8-2 exhibit significant antibacterial and antifungal activities against *Xanthomonas arboricola* pv. pruni (Xap) and *Mycosphaerella cerasella* (Mc), which cause shot-hole disease in flowering cherry trees [53]. The broad-spectrum antifungal potential of *B*. *subtilis* PBs 12 and *B*. *paralicheniformis* 36 was demonstrated via PCR screening for genes (spoVG, bacA, and srfAA AMP) encoding antimicrobial compounds such as subtilin, bacillysin, and surfactin [54]. Difficidin and bacilysin produced by *B. velezensis* FZB42 alter the cell wall structure of *Xanthomonas oryzae* pv. oryzae and *X*. *oryzae* pv. *oryzicola*, downregulating the expression of genes involved in *Xanthomonas* virulence, cell division, and protein and cell wall synthesis, thus reducing bacterial blight and leaf streak severity in rice [55]. Macrolides produced by *B. subtilis* BS-58 effectively inhibit *Fusarium oxysporum* and *Rhizoctonia solani*, pathogens in amaranth [56]. Thus, LSR7 may synthesize various unknown substances with promising antifungal potential.

Studies demonstrate a positive correlation between the activities of many defense-related enzymes in plants and the disease resistance index, indicating protective responses against pathogens [57]. These enzymes aid in metabolizing disease-resistant secondary metabolites such as lignin, phenolic compounds, and plant defense hormones [58,59]. They directly inhibit and kill pathogenic bacteria via reactive oxygen species metabolism, enhancing plant resistance to pathogens [60]. It has been reported that endophytic can induce antioxidant enzymes associated with plant defense against attack by plant pathogens [61]. Endophytic *Bacillus altitudinis* GTS-16 from holy basil can activate and induce the synthesis of Apx, SOD, PAL, and PO, which can induce systemic resistance against *Rhizoctonia solani* in rice [62]. In this study, the application of *B*. *velezensis* LSR7 significantly enhanced the activities of various defense-related enzymes in rubber tree plants, with a stronger effect at higher concentrations, similar to *B*. *velezensis* B4-7′s effect on tobacco seedlings [63]. Thus, LSR7 improved rubber tree resistance to red root rot by increasing the defense enzyme activity of rubber tree seedlings, suggesting that the biocontrol activity of LSR7 against *G*. *pseudoferreum* may be related to various mechanisms and their synergistic effects. 

## 5. Conclusions

The bacterial strain LSR7 was isolated from the roots of rubber trees and strongly inhibited the growth of *G. pseudoferreum*, the causative agent of red root rot in rubber trees. The LSR7 sequence was identical to that of *B. velezensis*. Dual-culture tests and greenhouse experiments indicated that LSR7 significantly inhibited *G. pseudoferreum* and promoted rubber tree growth. Strain LSR7 produces hydrolytic enzymes, siderophores, and active substances that inhibit *G. pseudoferreum* growth. Additionally, LSR7 enhanced defense enzyme activity and induced disease resistance in rubber trees. Therefore, LSR7 has the potential for preventing and treating red root rot disease in rubber trees. Further research is needed to uncover additional biocontrol mechanisms involved in the antimicrobial activity of LSR7 and identify the components of its antifungal compounds.

## Figures and Tables

**Figure 1 jof-10-00849-f001:**
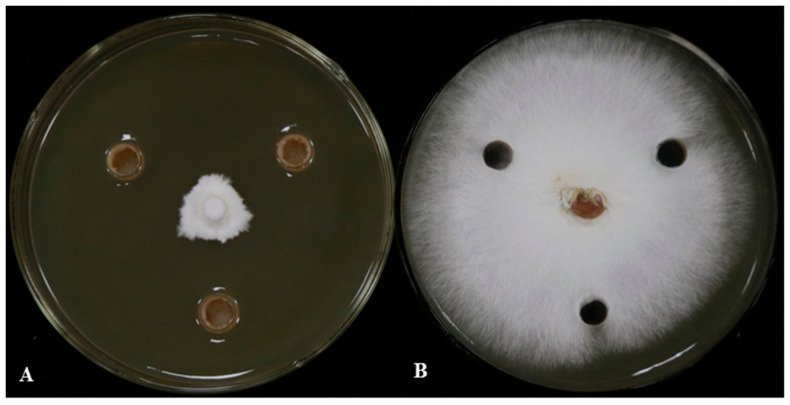
LSR7 antifungal activity against *G. pseudoferreum* (inhibition rate: 88.49%). Bacterial cultures (**A**) and sterile water (**B**) were inoculated on the three-point symmetry of a fungal plug.

**Figure 2 jof-10-00849-f002:**
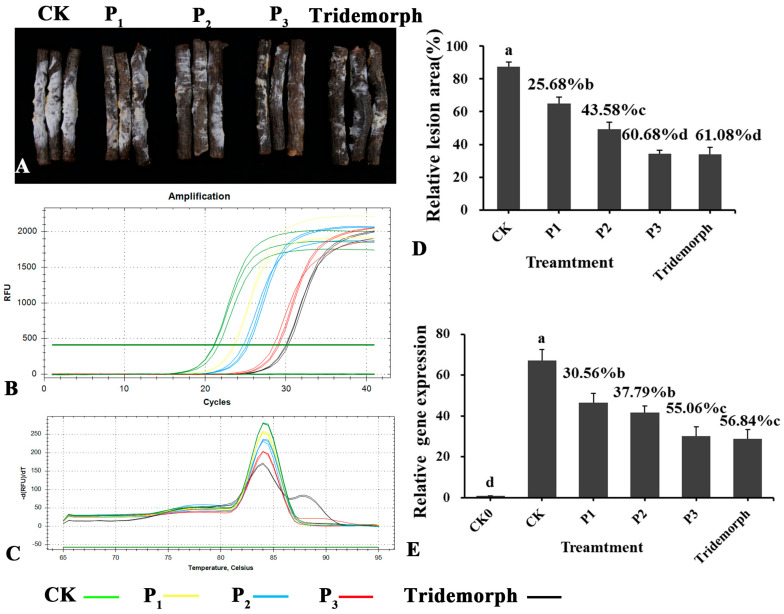
Evaluation of *B. velezensis* LSR7 for the biocontrol of red root rot in rubber trees. P_3_~P_1_: 10^8^~10^6^ CFU/mL of LSR7 + *G. pseudoferreum*, CK: NB + *G. pseudoferreum*, CK_0_: Only NB. Tridemorph: positive control. Biocontrol effectiveness of *B. velezensis* LSR7 on the roots of *Hevea brasiliensis* against red root rot under different treatments. RT-qPCR amplification curves (**A**) and melting curve (**C**) of *G. pseudoferreum* cDNA under different treatments. Biocontrol efficiencies of *B. velezensis* LSR7 were calculated according to the expression values of *ACT*. Infection of rubber tree by *G. pseudoferreum* under different treatments (**B**). The effect of different concentrations of LSR7 on red root rot disease of *Hevea brasiliensis* was calculated by means of relative lesion area (**D**) and relative gene expression (**E**). Numerical values represent the mean ± standard deviation of the triplicate experiments. Means were tested using Duncan’s multiple range test using the SPSS version 23 software. Means followed by the same letter within the same column are not significantly different (*p* < 0.05).

**Figure 3 jof-10-00849-f003:**
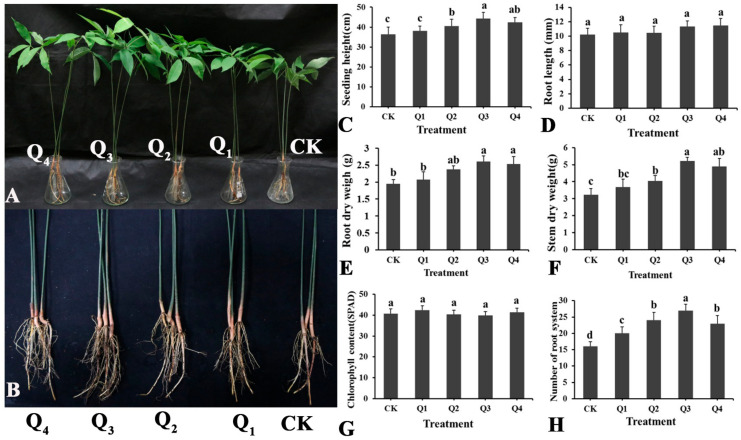
Effect of LSR7 on the rubber tree seedling growth (**A**,**B**) (n = 20). Effect of LSR7 on the rubber tree seedling height (**C**), root length (**D**), root dry weight (**E**), stem dry weight (**F**), chlorophyll content (**G**), and number of root systems (**H**). Bars indicate the standard error of the mean. Columns marked with the same letter are not significantly different at *p* < 0.05, according to Duncan’s multiple range test. Q_1~_Q_4_: 10^6^~10^9^ CFU/mL of LSR7, CK: NB.

**Figure 4 jof-10-00849-f004:**
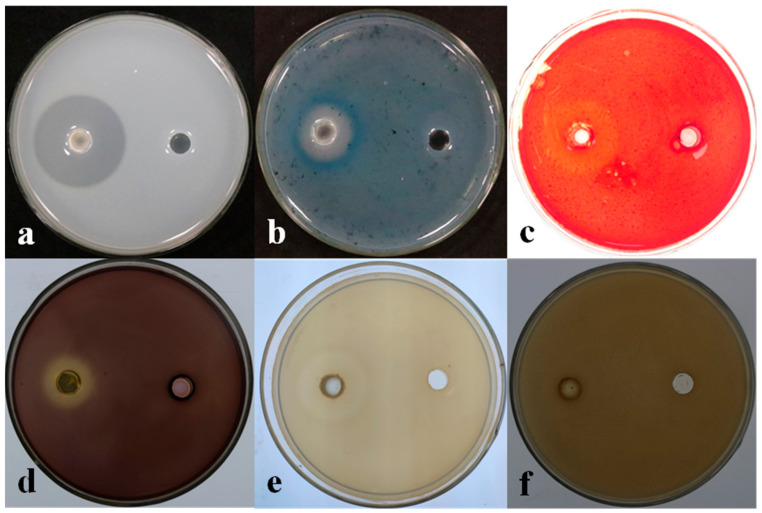
Detection of cell wall-degrading enzymes and siderophore (n = 15): (**a**). protease, (**b**). siderophore, (**c**). cellulase, (**d**). amylase, (**e**). β-1,3-glucanase, and (**f**). chitinase.

**Figure 5 jof-10-00849-f005:**
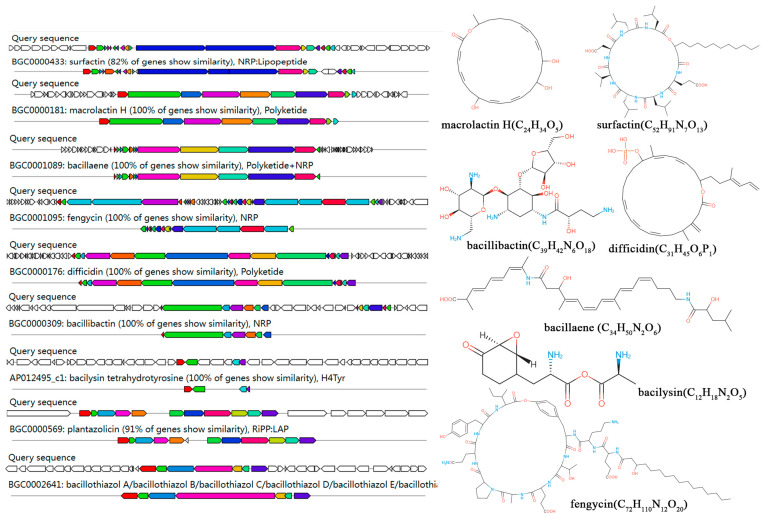
Genome annotation of LSR7 and prediction of biosynthesis gene clusters (BGCs). Genomic information and chemical structure of BGCs compared to those of known BGCs.

**Figure 6 jof-10-00849-f006:**
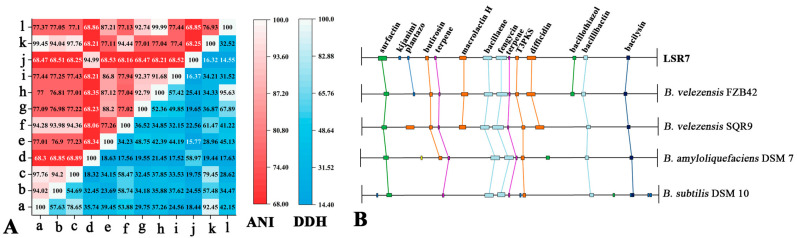
(**A**) ANIb and DDH values of LSR7 with 12 *Bacillus* strains. a. LSR7; b. *B. amyloliquefaciens* DSM7; c. *B. velezensis* FZB42; d. *B. cereus* ATCC 10987; e. *B. mojavensis* KCTC 3706; f. *B. siamensis* KCTC 13613; g. *B. spizizenii* ATCC 6633; h. *B. subtilis* ATCC 6051a; i. *B. tequilensis* KCTC 13622; j. *B. thuringiensis* LM1212; k. *B. velezensis* NJN-6; l. *B. subtilis* strain DSM 10. (**B**) Comparison of the secondary metabolite biosynthesis gene clusters of *B*. *velezensis* LSR7 with *B*. *velezensis* FZB42, *B*. *velezensis* SQR9, *B*. *amyloliquefaciens* DSM7, and *B*. *subtilis* DSM10.

**Figure 7 jof-10-00849-f007:**
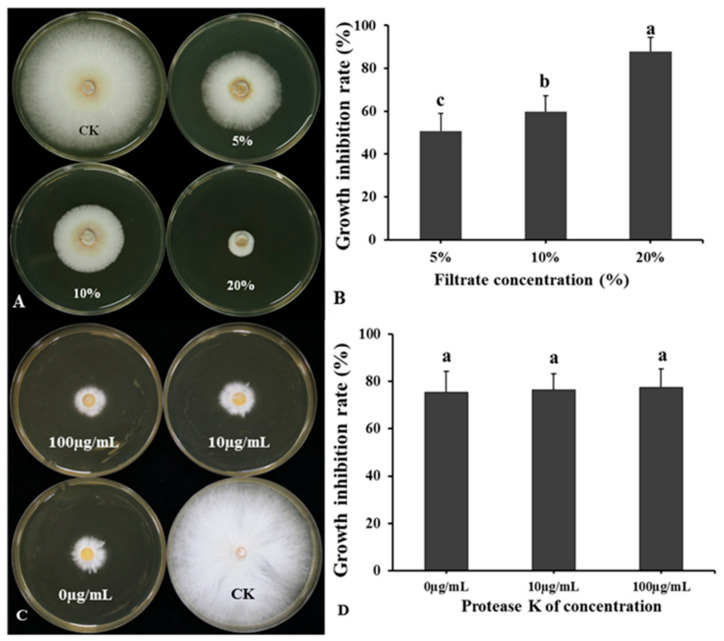
Effect of BCF amended to molten agar at different concentrations on the mycelial growth of *G. pseudoferreum* (**A**,**B**) (n = 15). Antifungal activity of BCF treated with different concentrations of protease K against *G. pseudoferreum* (**C**,**D**). Bars indicate the standard error of the mean. Columns marked with the same letter are not significantly different at *p* < 0.05, considering Duncan’s multiple range test (**B**,**D**) (n = 15).

**Figure 8 jof-10-00849-f008:**
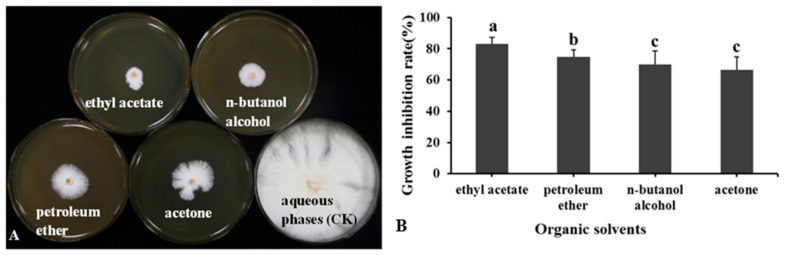
Antifungal activity of LSR7 lipopeptide metabolites extracted by different organic solvents against *G. pseudoferreum* (**A,B**) (n = 15). Bars indicate the standard error of the mean. Columns marked with the same letter are not significantly different at *p <* 0.05, considering Duncan’s multiple range test (**B**).

**Figure 9 jof-10-00849-f009:**
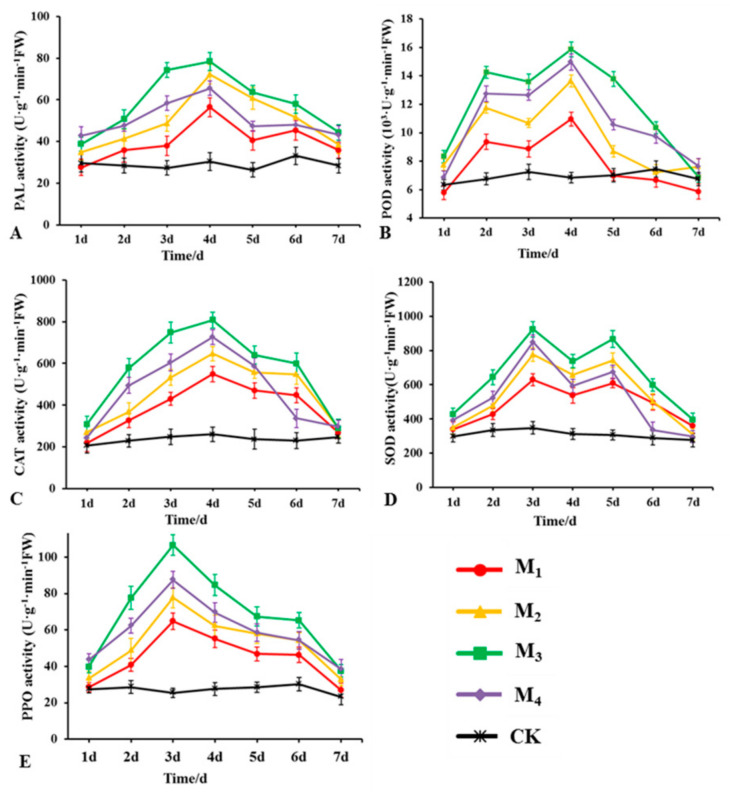
Changes in (**A**) PAL, (**B**) POD, (**C**) CAT, (**D**) SOD, and (**E**) PPO activities in rubber tree seedling roots after LSR7 treatment (n = 20). M_4_~M_1_: 10^9^~10^6^ CFU/mL of LSR7; CK: Hoagland’s nutrient solution without LSR7.

**Table 1 jof-10-00849-t001:** Inhibitory activity of bacterial endophytes from rubber tree against *G. pseudoferreum* (n = 15).

Strain	Antifungal Bandwidth (mm)	Inhibition Rate (%)
WSR6	10.54 ± 0.52	72.32 ± 1.11 b
LSR7	13.86 ± 0.45	81.06 ± 2.68 a
DHS35	7.67 ± 0.55	66.67 ± 1.33 c
DHR46	8.99 ± 0.72	69.85 ± 1.74 c
DHR27	12.92 ± 0.32	79.33 ± 0.78 a
BHS5	10.85 ± 0.33	74.33 ± 0.80 b
QSS3	8.62 ± 0.62	68.96 ± 1.49 c
QHL1	12.94 ± 0.45	79.38 ± 1.08 a

Numerical values are mean ± SD of triplicates. Means were tested using Duncan’s multiple range test of the SPSS version 23 software. Means followed by the same letter are not significantly different (*p* < 0.05) within the same line.

## Data Availability

The original contributions presented in the study are included in the article/Appendix A, and further enquiries can be directed to the corresponding authors. The LSR7 genome data presented in this study are available in the National Centre for Biotechnology Information (NCBI) database (GenBank number: JBGTYF000000000.1) at the NCBI database.

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
