# Peer review of "Biocontrol Potential of Endophytic Bacillus velezensis LSR7 Against Rubber Red Root Rot Disease"

_jof, 2024, doi:10.3390/jof10120849_

Round 1

Reviewer 1 Report

This report described the process how a bacterial strain was isolated, screened for its antifungal ability and characterized for its mechanisms. Very nice work.

1, Ganoderma is a fungus, however, bactericide was used many time in the paper regarding the biocontrol activity of the BCA against the pathogen. This need to be corrected throughout the paper. Fungicide is a better term.

2, l21-27, the identification process of the BCA need to be mentioned.

3, L68, nitrogen fixation is a process, not a chemical. Similarly, Volatile organic compounds are not endophytes.

4, L77, endophytes are only one group of the BCAs, and there is no clear evidence to support your statement.

5, is there information about biocontrol of the disease with other types of BCAs?

6, l100, what media was used in plating the bacteria?

7, l103-106, this procedure seems that you are not confident with your surface sterilization technique, or you can cite a reference to support your technique.

8, l08-109, need reference for this technique or need to mention how you tested it.

9, l121, what was used as control?

10, L125-127, you only sequenced on strain, but you were talking about many strains this way.

11, L137-150, did you make the wound before or after bacterial inoculation? Is wound critical for the pathogen to infect the plant root?  I did not see this in the Introduction.

12, l153-160, what gene was used to quantify the pathogen?

13, l294-298, rewrite the section for clarification. L299-300, what was the result of strain identification from the culture collection center since your identification was through sequence comparison? Did the center conduct other studies?

14, l327-330, the increase of seedling growth was determined by comparing to what?

Author Response

Comments 1: Ganoderma is a fungus, however, bactericide was used many time in the paper regarding the biocontrol activity of the BCA against the pathogen. This need to be corrected throughout the paper. Fungicide is a better term.

Response 1: Thank you very much. The full paper has been revised according to your comments.

Comments 2: L21-27, the identification process of the BCA need to be mentioned.

Response 2: This section has been added. L19-20

Comments 3: L68, nitrogen fixation is a process, not a chemical. Similarly, Volatile organic compounds are not endophytes.

Response 3: We have made changes. What we mean is that volatile organic compounds are produced by endophytes. L68, L71

Comments 4, 5: L77, endophytes are only one group of the BCAs, and there is no clear evidence to support your statement. Is there information about biocontrol of the disease with other types of BCAs?

Response 4, 5: Thank you for your opinion, endophytic bacteria is only one group of BCAs, BCAs also includes plant rhizosphere bacteria, water bacteria and so on. So we modified that statement. L78-79

Comments 6: L100, what media was used in plating the bacteria?

Response 6: The plant tissue was ground with sterile mortar, the tissue liquid collected after grinding was diluted 10-fold with sterile water and plating in nutrient agar (NA) plates. L102-104

Comments 7: L103-106, this procedure seems that you are not confident with your surface sterilization technique, or you can cite a reference to support your technique.

Response 7: We are confident in our technology and references have been cited in the manuscript.

Comments 8: Ll08-109, need reference for this technique or need to mention how you tested it.

Response 8: We have added in the manuscript. L110

Comments 9: L121, what was used as control?

Response 9: “Plates inoculated with equal amounts of NB were used as controls”. We have added in the manuscript. L123

Comments 10: L125-127, you only sequenced on strain, but you were talking about many strains this way.

Response 10: We have modified this part of the content. L133

Comments 11: L137-150, did you make the wound before or after bacterial inoculation? Is wound

critical for the pathogen to infect the plant root? I did not see this in the Introduction.

Response 11: We made the wound after we inoculated it with bacteria. Even if no wound pathogens can infect the root segment, but in the experiment, creating wounds is conducive to pathogen infection.

Comments 12: L153-160, what gene was used to quantify the pathogen?

Response 12: The 18S rRNA gene was the internal reference, and ACT gene was the target gene of pathogen Ganoderma pseudoferreum. This section has been added in the manuscript. L165

Comments 13: L294-298, rewrite the section for clarification. L299-300, what was the result of strain identification from the culture collection center since your identification was through

sequence comparison? Did the center conduct other studies?

Response 13: It has been modified and supplemented. The strain Culture collection center determined the classification status of strain LSR7 by detecting strain morphological characteristics, 16S rRNA gene sequence, physiological and biochemical characteristics and whole genome detection. The results identified by the culture collection center were consistent with ours, that is, the strain LSR7 was B. velezensis.L304-319

Comments 14: L327-330, the increase of seedling growth was determined by comparing to what?

Response 14: (Growth parameters of treatment group - growth parameters of control group)/Growth parameters of control group ×100%.

Reviewer 2 Report

In the Abstract section, simplify the introductory paragraph, clearly specify the aims of the work, the methodology, and improve the description of the results.

Line 16. I suggest that the text should be modified and should be written as follows, “…Long-term use of chemical fungicides…”

Line 18. I suggest that the text should be modified and should be written as follows, “…The Bacillus velezensis strain LSR7, isolated…”

It is suggested that the keywords be literally different from those in the title of the work.

Lines 83-89. The aims must be reviewed and corrected, this is because they only focused on and show results for Bacillus velezensis.

Line 100. “…for 10-fold dilution and plating…” In which culture medium?

Line 108. “…Endophytic bacteria with antagonistic effects against G. pseudoferreum…” How many bacterial isolates did they obtain? Authors should specify this information in this section.

Line 148. “…placed in a humid box and cultured at 28 °C for 7 d…” ¿ What was the photoperiod? Or was the incubation in total darkness?

Line 154. “…Total RNA was extracted from the…” Authors are encouraged to specify why they did not include a DNase digestion.

Line 157. “…via qPCR using ACT and 18S rRNA as the internal…” According to your results, the 18S rRNA gene was the internal reference, and ACT was the target gene, is this correct? Also, if you used SYBR qPCR Master Mix for the qPCR, why didn't you include a melting curve analysis? You should clearly specify this in this section. Also see doi:10.1038/nprot.2008.73

Line 160. “…expression levels were calculated using the 2−ΔΔCT method…” It is understandable that the 18S rRNA gene was the internal reference, but what was the calibrator treatment? The 2−ΔΔCT value of this treatment is usually 1.

In the methodology section, in those bioassays where applicable, you must specify which experimental design was used in the establishment.

Line 240. Antibacterial activity or Antifungal activity?

Line 282. What was the protocol for determining enzyme activity?

Line 286. Was an ANOVA performed? It is observed that you present percentage data for some variables. Why were statistical transformations not performed?

Line 289. Let's start by briefly describing the results of the “Isolation and screening of antagonistic bacteria from rubber tree”

Line 292. “…inhibition rate of 88.49%...” Is this a mean value? You should also mention the standard error. n=?

Line 302. It would be appropriate to also present results from some other isolates here for comparative purposes.

Line 308. “…were 25.68%, 43.58%, and 60.67%...” Are these mean values ​? Should you also mention the standard error for each case? n=? Do the same for lines of text 310-313.

Lines 316-321. In the figure caption you must describe the meaning of the treatments, CK, P1, P2, P3, y Tridemorph.

Line 315. When using the 2−ΔΔCT method, usually the Control group is the calibrator and has a value of 1. According to your results in Figures 2B and 2D, neither CK nor Tridemorph were the calibrators in determining relative mRNA levels. What happened here?

Lines 336-339. In the figure caption you must describe the meaning of the treatments CK, Q1, Q2, Q3, y Q4. n=?

Line 398. “…were 50.61, 59.86, and 87.86%, respectively…” Are these mean values ​? Should you also mention the standard error for each case? n=?

Lines 395 and 414. “Effect of BCF on mycelial growth of G. pseudoferreum” Are they the same? Or why the same subheading?

Line 418. Are these mean values ​? Should you also mention the standard error for each case? n=? Why the antibacterial activity?

Line 422. Is G. pseudoferreum a bacterium or why the antibacterial activity?

Line 446. Why didn't they perform the DMRT test here?

Line 463. The antibacterial mechanism?

Lines 526-530. X.-J.M., and H.-B.C. Si these authors contributed equally to this study, Why is this not seen in the Author Contributions?

There are spelling and grammatical errors throughout the manuscript that need to be reviewed. The Discussion section seems to be very brief, it is suggested that they improve it, and they should separate the justification of each variable evaluated by means of subheading, as they did in the Results section. It is evident that they are not interpreting-discussing all the variables evaluated in the experiments carried out.

Author Response

Comments 1: In the Abstract section, simplify the introductory paragraph, clearly specify the aims of the work, the methodology, and improve the description of the results.

Response 1: Thank you for your suggestion, we have modified according to your comments.

Comments 2: Line 16. I suggest that the text should be modified and should be written as follows,

“…Long-term use of chemical fungicides…

Response 2: Thank you for your suggestion, we have modified according to your comments. Line16.

Comments 3: Line 18. I suggest that the text should be modified and should be written as follows,

“…The Bacillus velezensis strain LSR7, isolated…”

Response 3: Thank you for your suggestion, when we isolated LSR7, we did not know it was Bacillus velezensis, so we do not think it is appropriate to write this here. But we have also made changes based on the advice of other experts. Line19-20

Comments 4: It is suggested that the keywords be literally different from those in the title of the work.

Response 4: Thank you for your suggestion, we have modified according to your comments.

Comments 5: The aims must be reviewed and corrected, this is because they only focused on and show results for Bacillus velezensis.

Response 5: Thank you for your questions. We had not conducted experiments at the time of determining the target, so we did not know that the highly effective antagonistic bacteria that would be obtained in the future would be Bacillus velezensis.

Comments 6: Line 100. “…for 10-fold dilution and plating…” In which culture medium?

Response 6: The plant tissue was ground with sterile mortar, the tissue liquid collected after grinding was diluted 10-fold with sterile water and plating in nutrient agar (NA) plates. Line 102-104

Comments 7: Line 108. “…Endophytic bacteria with antagonistic effects against G. pseudoferreum…” How many bacterial isolates did they obtain? Authors should specify this information in this section.

Response 7: We have added your suggestions in the results section. Line 303

Comments 8: Line 148. “…placed in a humid box and cultured at 28 ℃ for 7 d…”? What was the photoperiod? Or was the incubation in total darkness?

Response 8: It was hatched in total darkness. We have added in the manuscript. Line 155

Comments 9: Line 154 “…Total RNA was extracted from the…” Authors are encouraged to specify why they did not include a DNase digestion.

Response 9: Total RNA was isolated from the mycelia of G. pseudoferreum on each wound in the root segments. We used DNase to digestion gDNA during the extraction of total RNA. Line 161

Comments 10: Line 157. “…via qPCR using ACT and 18S rRNA as the internal…” According to your results, the 18S rRNA gene was the internal reference, and ACT was the target gene, is this correct? Also, if you used SYBR qPCR Master Mix for the qPCR, why didn't you include a melting curve analysis? You should clearly specify this in this section. Also see doi:10.1038/nprot.2008.73.

Response 10: the 18S rRNA gene was the internal reference, and ACT was the target gene, it is correct. Thank you for your literature. We believe that the melting curve is mainly to determine the specificity of the primer, and the specificity of our primer meets the research requirements, which has been confirmed in previous studies. https://doi.org/10.3390/microorganisms12091793.

Comments 11: Line 160. “…expression levels were calculated using the 2−ΔΔCT method…” It is understandable that the 18S rRNA gene was the internal reference, but what was the calibrator treatment? The 2−ΔΔCT value of this treatment is usually 1.

Response 11: In this experiment, we have not inoculated the root of pathogen and LSR7 as the calibrator treatment. We explain in the manuscript. Line170.

Comments 12: In the methodology section, in those bioassays where applicable, you must specify which experimental design was used in the establishment.

Response 12: We have made corrections.

Comments 13: Line 240. Antibacterial activity or Antifungal activity?

Response 13: Antifungal activity. We have modified it.

Comments 14: Line 282. What was the protocol for determining enzyme activity?

Response 14: Enzyme activity is determined according to the instructions in the kit. We have added in the manuscript. Line 297

Comments 15: Line 286. Was an ANOVA performed? It is observed that you present percentage data for some variables. Why were statistical transformations not performed?

Response 15: We performed ANOVA for all variables data are provided in the resulting figures and tables.

Comments 16: Line 289. Let's start by briefly describing the results of the “Isolation and screening of antagonistic bacteria from rubber tree”.

Response 16: We have already added this part. Line 304

Comments 17: Line 292. “…inhibition rate of 88.49%...” Is this a mean value? You should also mention the standard error. n=?

Response 17: This is the average because we repeated the experiment for each treatment. We have supplemented this data table in the results section. n=15. Line 322

Comments 18: It would be appropriate to also present results from some other isolates here for comparative purposes.

Response 18: We have supplemented this data table in the results section. Line 322

Comments 19: Line 308. “…were 25.68%, 43.58%, and 60.67%...” Are these mean values? Should you also mention the standard error for each case? n=? Do the same for lines of text 310-313.

Response 19: These are mean values. We have marked the standard error in the figure, and in order to reflect the standard error more directly, we have marked the standard error again in the text of the result. n=20. Line 330-337

Comments 20: Lines 316-321. In the figure caption you must describe the meaning of the treatments, CK, P1, P2, P3, y Tridemorph.

Response 20: We have already added this part. Line 338, Line 343

Comments 21: Line 315. When using the 2−ΔΔCT method, usually the Control group is the calibrator and has a value of 1. According to your results in Figures 2B and 2D, neither CK nor Tridemorph were the calibrators in determining relative mRNA levels. What happened here?

Response 21: Thank you for this question. In this experiment, we have not inoculated the root of pathogen and LSR7 as the calibrator treatment. We explain in the manuscript. Line170.

Comments 22: Lines 336-339. In the figure caption you must describe the meaning of the treatments CK, Q1, Q2, Q3, y Q4. n=?

Response 22: We have already added this part. n=20. Line 366.

Comments 23: Line 398. “…were 50.61, 59.86, and 87.86%, respectively…” Are these mean values? Should you also mention the standard error for each case? n=?

Response 23: These are mean values. We have marked the standard error in the figure, and in order to reflect the standard error more directly, we have marked the standard error again in the text of the result. n=15. Line 423

Comments 24:Lines 395 and 414. “Effect of BCF on mycelial growth of G. pseudoferreum” Are they the same? Or why the same subheading?

Response 24: This was our mistake and we have corrected it. Lines 420 and 440

Comments 25: Line 418. Are these mean values? Should you also mention the standard error for each case? n=? Why the antibacterial activity?

Response 25: These are mean values. We have marked the standard error in the figure, and in order to reflect the standard error more directly, we have marked the standard error again in the text of the result. n=15. Antifungal activity. We have modified it. Line 443

Comments 26: Line 422. Is G. pseudoferreum a bacterium or why the antibacterial activity?

Response 26: We have modified it. “Antibacterial”should be “antifungal”

Comments 27: Line 446. Why didn't they perform the DMRT test here?

Response 27: We visually displayed the trend of enzyme activity change over time in each treatment group by means of line charts. If DMRT test was carried out, the data of each treatment group at each time should be marked with significant differences, which would make the picture very confusing. This was the case in previous studies.

https://doi.org/10.1016/j.biocontrol.2021.104785

Comments 28: Line 463. The antibacterial mechanism?

Response 28: Antifungal mechanism. We have modified it.

Comments 29: Lines 526-530. X.-J.M., and H.-B.C. Si these authors contributed equally to this study, Why is this not seen in the Author Contributions.

Response 29: Thank you for your comments.This manuscript tribute to the same author is the conclusion of our comprehensive consideration. Even if the contributions are the same, it is impossible for two authors to have exactly the same work tasks. For example, the sources of funding are not available to everyone.

Comments 30: There are spelling and grammatical errors throughout the manuscript that need to be reviewed. The Discussion section seems to be very brief, it is suggested that they improve it, and they should separate the justification of each variable evaluated by means of subheading, as they did in the Results section. It is evident that they are not interpreting-discussing all the variables evaluated in the experiments carried out.

Response 30:Thank you for your comments. We have made appropriate changes based on your comments.

Reviewer 3 Report

The article provides relevant information on the biological control of red rot in rubber trees using B. velezensis. The study is robust and evaluates control across various aspects, both in vivo and in vitro, as well as the mechanisms involved. Authors must pay attention to the journal's standards and the correct writing of scientific concepts.

Some adjustments were requested in the methodology.

Author Response

Comments 1: Wouldn't the more current name be Ganoderma philippii?

Response 1: Ganoderma pseudoferreum and Ganoderma philippii are considered synonyms in some literature. Ganoderma pseudoferreum has been commonly used as the pathogen of red root rot of rubber tree reported in China, and Ganoderma philippii has been widely reported in Malaysia, Indonesia and the Philippines, so we use the name Ganoderma pseudoferreum.

https://repository.naturalis.nl/pub/531977/; http://www.tandfonline.com/loi/tsfs20

Comments 2: Some of these words are already in the title!

Response 2: We made appropriate modifications.

Comments 3: As? Spectrophotometer?

Response 3: Yes, we used a spectrophotometer. (OD600 ≈108CFU/ml)

Comments 4: Italic problem

Response 4: We have modified it.

Round 2

Reviewer 2 Report

Authors are encouraged to respond to comments and suggestions, with the aim of ensuring that their research results have the relevant scientific rigor.

In the Abstract section, clearly specify the aims of the work, the methodology, and improve the description of the results.

It is suggested that the keywords be literally different from those in the title of the work.

Line 161. Authors should specify the inclusion of a DNase digestion.

Line 168. Authors should specify the inclusion of a melting curve analysis

Line 160. “…expression levels were calculated using the 2−ΔΔCT method…” It is understandable that the 18S rRNA gene was the internal reference, but what was the calibrator treatment? The 2−ΔΔCT value of this treatment is usually 1.

Lines 168-171. The explanation is confusing, the authors are encouraged to improve their description and to review https://doi.org/10.1038/nprot.2008.73

In methodology section, in those bioassays where applicable, you must specify which experimental design was used in the establishment.

Line 299. Authors should specify whether an ANOVA was performed, and explain why they did not perform prior statistical transformations; for percentage data this is highly advisable.

In Results section, authors are encouraged to specify n values ​​in figure or table captions.

Line 325. Regarding the DMRT test, in Table 1, which letters do they refer to?

Author Response

Comments 1: In the Abstract section, clearly specify the aims of the work, the methodology, and improve the description of the results.

Response 1: We have made changes to the abstract section

Comments 2: It is suggested that the keywords be literally different from those in the title of the work.

Response 2: Thank you for your suggestion, but we are having a hard time finding keywords that match the content of our manuscript

Comments 3: Line 161. Authors should specify the inclusion of a DNase digestion.

Response 3: We have added it in the Materials and methods section. Line 162

Comments 4: Line 168. Authors should specify the inclusion of a melting curve analysis.

Response 4: We have added it in the Materials and methods section. Line 342

Comments 5: Line 160. “…expression levels were calculated using the 2−ΔΔCT method…” It is understandable that the 18S rRNA gene was the internal reference, but what was the calibrator treatment? The 2−ΔΔCT value of this treatment is usually 1.

Response 5: We have made revisions in the manuscript. Line 342 (Figure 2), Line 170

Comments 6: Lines 168-171. The explanation is confusing, the authors are encouraged to improve their description and to review https://doi.org/10.1038/nprot.2008.73

Response 6: We have revised your request accordingly. Line 342 (Figure 2), Line 170

Comments 7: In methodology section, in those bioassays where applicable, you must specify which experimental design was used in the establishment.

Response 7: We have revised your request accordingly, and if we have not yet reached your request, we would like you to clearly specify the problem location.

Comments 8: Line 299. Authors should specify whether an ANOVA was performed, and explain why they did not perform prior statistical transformations; for percentage data this is highly advisable.

Response 8: We have carried out ANOVA on the relevant results (Line 300), which may be because we did not make it clear in the manuscript, and now we have modified it in the corresponding position. We have done the same statistical transformations before, but it is only indicated in the figure and table in the result section, not in the text section.

Comments 9: In Results section, authors are encouraged to specify n values ​​in figure or table captions.

Response 9: Thanks for your suggestion, we have added it in the results section.

Comments 10: Line 325. Regarding the DMRT test, in Table 1, which letters do they refer to?

Response 10: The last missing letters were our fault. We did ANOVA and significance difference analysis. Now we have added the letters. Line 325.